# CrayonRobo: Toward Generic Robot Manipulation via Crayon Visual Prompting

## Abstract

In robotic manipulation, there are several ways to convey the task goal, including language conditions, goal images, and goal videos. However, natural language can be ambiguous, and images or videos can be over-specified. To address this issue, we propose an innovative approach using a straightforward and practical representation: crayon visual prompts, which explicitly indicate both low-level actions and high-level planning. Specifically, for each atomic step, our method allows drawing simple yet expressive 2D visual prompts on RGB images to represent the required actions, *i.e.*, end-effector pose and moving direction. We devise a training strategy that enables the model to comprehend each color prompt and predict the contact pose along with the movement direction in SE(3) space. Furthermore, we design an interaction strategy that leverages the predicted movement direction to form a trajectory connecting the sequence of atomic steps, thereby completing the long-horizon task. Through introducing simple human drawn prompts or automatically generated alternatives, we enable the model to explicitly understand its task objective and boost its generalization ability on unseen tasks by providing model-understandable crayon visual prompts. We evaluate our method in both simulation and real-world environments, demonstrating its promising performance.

## 1 Introduction

As thoughtful helpers for humans, it is crucial for robots to understand and successfully execute their assigned tasks. Various approaches exist to convey the goals that robots should achieve, such as language descriptions, goal images, or goal videos. Language instructions (Liang et al., 2023; Ahn et al., 2022; Nair et al., 2022; Lynch & Sermanet, 2020; Huang et al., 2023; Shridhar et al., 2022; 2023; Li et al., 2024; Yang et al., 2023a) can be ambiguous and brief, making it challenging for the robot to understand the tasks, or they can be overly detailed, increasing the difficulty for the model to follow. Goal images (Zhong et al., 2023; Black et al., 2023; Bousmalis et al., 2023; Lynch et al., 2020; Jiang et al., 2022), while providing an accurate target, often contain extraneous information irrelevant to the task, such as background elements and non-interactive objects. Some methods use human demonstration videos (Chane-Sane et al., 2023) or generated videos (Black et al., 2023; Du et al., 2023; Yang et al., 2023b; Du et al., 2024) to outline tasks step-by-step. However, human demonstration videos are burdensome to encode, and generated videos depend heavily on their quality. To address these challenges, several works propose using visual prompts as a convenient yet expressive modality for goal specification. These visual prompts are easy for users to create and can effectively convey the precise goals that the policy model should focus on.

Among visual prompt-conditioned approaches, methods have attempted to convey task goals more effectively. The RT-Sketch (Sundaresan et al., 2023) method emphasizes drawing the target state of the most relevant object to represent the goal. However, it only depicts the final state and overlooks the intermediate key frames that are crucial for successful task execution. In contrast, methods like RT-Trajectory (Nasiriany et al., 2024; Gu et al., 2023; Stone et al., 2023) illustrate the entire movement path of the end-effector, which helps bridge the gap between task components and enhances generalization. Despite its advantages, they provide only positional information and neglect the directional information of the end-effector, which is also critical for accurate task completion. Additionally, as trajectories become longer and overlap between atomic tasks, they can create confusion for the model regarding overall task planning.

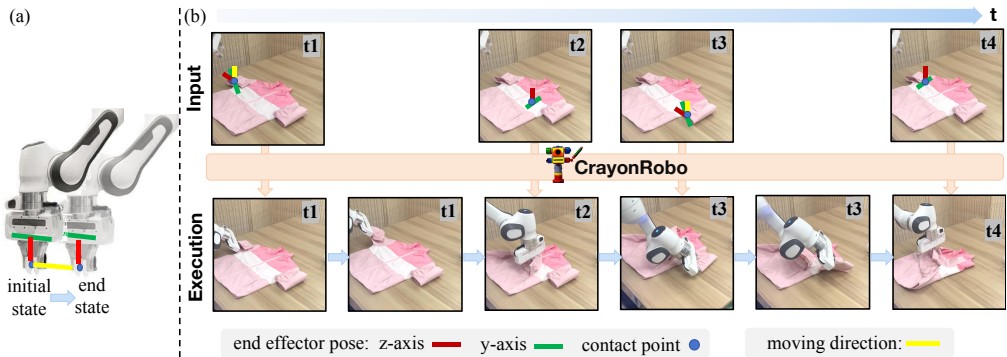

Figure 1: (a) The expression of different color prompts. (b) We utilize a sequence of images with crayon visual prompts to express the planning steps, with each step illustrating the required low-level atomic actions, *i.e.*, t1-pick, t2-place, t3-pick, t4-place. For simple tasks, such as t2-place, there is no need to draw the moving direction. Based on the input prompt, the model determines the 6DoF contact pose, enabling it to interact with the object as required. When a yellow prompt is present in the image, the model also predicts 3D movement directions, guiding the movement after contact, *e.g.*, picking upward in t1. By sequentially executing each step in the input sequence, the overall task is completed.

This leads us to consider: is there any way to precisely and non-redundantly convey the goal while also clearly communicating the end-effector's action? Therefore, we propose drawing crayon visual prompts on images to represent both low-level actions and high-level planning. Each step in the task is illustrated with 2D colored visual prompts that are simple, eliminating the need for task descriptions. As depicted in Figure 1(a), these prompts can include contact point, end-effector z-axis direction, end-effector y-axis direction, and moving direction after contact. Observing constraints among directions, such as the relationship between the corresponding 2D directional prompts and 3D directions, we develop a training strategy and supervision objectives aimed at progressively enhancing the model's ability to comprehend each color prompt. This allows the model to predict the corresponding SE(3) contact pose and the 3D movement direction, facilitating the completion of each atomic step. Furthermore, as illustrated in Figure 1(b), we use image sequences as input to express the task planning procedure, with each image drawn with crayon visual prompts indicating its atomic action goal. We design an interaction strategy that leverages the predicted 3D movement direction to connect each step sequentially, thereby formulating the overall trajectory. Through such simple visual prompts, we clearly instruct the model's action objectives while also enhancing its generalization capacity when encountering novel tasks by providing model-understandable visual prompts.

Our experimental setup encompasses a diverse range of manipulation tasks involving both familiar and novel tasks. Experiments show that our method can follow the drawn 2D instructional prompts or automatically generated alternatives and predict accurate 3D poses accordingly, achieving a promising manipulation success rate. In real-world scenarios, we validate the performance of our method on tasks that may not have been encountered during training. More demonstrations can be found in the supplementary video or at https://sites.google.com/view/crayonrobo.

In summary, our contributions are as follows:

- 1) We propose employing crayon visual prompts to explicitly convey the task objectives in both low-level action and high-level planning.

- 2) We train a model that comprehends the 2D prompts and predicts accurate contact poses along with moving directions in SE(3) space, ensuring the reliable completion of each atomic task and the connection between steps to formulate a long-horizon trajectory.

- 3) Experiments at scale demonstrate its promising performance and generalization ability.

## 2 RELATED WORK

*Pure vision-based manipulation.* In vision-based robotic manipulation, numerous studies have employed a variety of solutions, including deep learning (Brohan et al., 2022; Goyal et al., 2023; Shridhar et al., 2023; Brohan et al., 2023), imitation learning (Chi et al., 2023; Ze et al., 2024; Ke et al., 2024; Ju et al., 2024), and reinforcement learning (An et al., 2024; Luo et al., 2024; Dai et al., 2023; Nguyen & La, 2019). For example, in deep learning-based solutions, some methods design action policy networks (Mo et al., 2021; Eisner et al., 2022; Xu et al., 2022; Wen et al., 2023; Bahl et al., 2023) to calculate dense affordance maps and determine contact points and action poses. However, pure vision-based manipulation networks focus more on actionability than functionality, potentially failing when functional actions are required to meet human needs. To address this, more goal-oriented methods have been developed.

*Language-conditioned manipulation.* With advancements in language foundation models, language instructions are increasingly used for goal specification in goal-conditioned policy learning, as seen in (Brohan et al., 2022). Other works, such as (Liang et al., 2023; Ahn et al., 2022; Nair et al., 2022; Lynch & Sermanet, 2020; Huang et al., 2023; Shridhar et al., 2022; 2023; Li et al., 2024; Yang et al., 2023a; Xiao et al., 2022), utilize templated or freeform language for task specification. Building on this, (Belkhale et al., 2024) introduces language motions as an intermediate layer between high-level goals and low-level actions. Despite progress, challenges remain: language instructions often struggle to specify detailed actions or convey spatial objectives, frequently requiring human assistance to work effectively.

*Goal image or video-conditioned manipulation.* To enhance goal specificity and detail, several image-conditioned policy representations have been developed, with goal-image conditioning being one of the most prominent techniques. In goal-image conditioning, a final goal image defines the desired end state of a task (Zhong et al., 2023; Black et al., 2023; Bousmalis et al., 2023; Lynch et al., 2020; Jiang et al., 2022). This approach inputs both the initial and target states of the object and outputs the actions needed to achieve the goal. However, using goal images can be problematic because they provide a strong prior with excessive information, much of which may be irrelevant to the task, such as background details or unrelated objects. In addition to static images, some methods use video or generated video (Black et al., 2023; Du et al., 2023; Yang et al., 2023b; Du et al., 2024) to represent each step of the process frame by frame, offering a more detailed execution procedure. While this approach provides both high-level planning and low-level action details, encoding long videos can place a significant burden on the model.

*Visual prompt-conditioned manipulation.* To address the issue of redundant goal information, recent approaches have proposed using visual prompts as goals. These prompts provide concise, task-relevant information, which helps reduce unnecessary complexity and enhance task performance. For instance, (Sundaresan et al., 2023) suggests using goal sketches to indicate the target of the task-involved object, while (Gu et al., 2023) proposes drawing moving trajectories with key waypoints to specify the desired movement of the end-effector. Additionally, (Nasiriany et al., 2024) introduces a method that iteratively selects waypoints from a pool of potential options to form the overall trajectory, and (Yang et al., 2023a; Stone et al., 2023; Liu et al., 2024a) use external models to choose waypoints based on mark-based visual prompts to complete tasks. While these approaches effectively focus on the robot's position or movement, they often overlook the importance of rotation, which is also crucial for task execution. Moreover, some methods consider the task as a whole, leading to potential visual prompt overlap as the task lengthens.

Based on these insights, we propose an innovative solution that allows drawing crayon visual prompts to represent both low-level action and high-level planning goals. This approach clearly conveys the objective of each individual step, *i.e.*, where and how to interact with the object, as well as the required task procedure.

## 3 METHOD

### 3.1 PROBLEM FORMULATION

The task is defined as follows: to start with, the model takes the visual input $I \in \mathbb{R}^{H \times W \times 3}$ of an object drawn with crayon visual prompts. This prompt utilizes four colors: blue, representing the

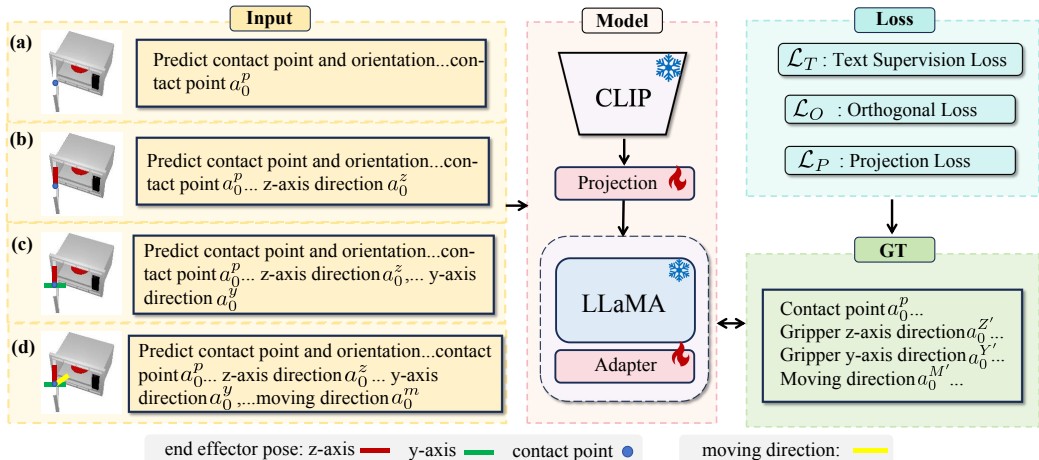

Figure 2: We design training pairs that convey varying levels of information to enable the model to comprehend each visual prompt and introduce loss functions to guide it in predicting accurate poses.

contact point; red and green, indicating the z-axis and y-axis directions of the end-effector when contacting the object; and yellow, symbolizing the moving direction after contacting. To eliminate ambiguities caused by visual prompts overlapping each other, we specify their numerical values in text inputs as a language prompt $P$, including the coordinates of the contact point denoted as $a_0^p \in \mathbb{R}^2$, the depicted gripper's 2D z-axis direction $a_0^z$ and y-axis direction $a_0^y$, and the 2D moving direction $a_0^m$, all represented as unit vectors in $\mathbb{R}^2$. The objective of the model is to generate an action $a_0 = (a_0^p, a_0^Z, a_0^Y, a_0^M)$, where $a_0^p$ is then mapped into 3D action space $a_0^P$ using the depth information of the contact point, while $a_0^Z$, $a_0^Y$, and $a_0^M$ are unit vectors in $\mathbb{R}^3$ representing the end-effector's 3D z-axis, y-axis, and moving directions, respectively. $a_0^Z$ and $a_0^Y$ jointly determine the rotation for contacting the object, while $a_0^M$ controls the following movement.

## 3.2 DATA COLLECTION

In this section, we elaborate on how to obtain the crayon visual prompt required for training. In the simulator, we interact randomly with objects. If successful manipulation occurs after a random interaction, we record the success contact pose of the end-effector including 3D contact point $a_0^{P'}$, z-axis direction $a_0^{Z'}$, and y-axis direction $a_0^{Y'}$ along with the moving direction $a_0^{M'}$ of the object contact point. These are then served as ground-truth to guide the training. $a_0^{P'}$ is projected onto a 2D image as $a_0^p = (x, y)$ using camera parameters, appearing as the blue dots. As for the directional visual prompt, we adopt $a_0^{P'}$ as the center and locate other 3D points along $a_0^{Z'}$ and $a_0^{Y'}$. These points are then projected onto the 2D image. Subsequently, we connect these points with $a_0^p$ to calculate 2D vectors $a_0^z$ and $a_0^y$, while drawing red and green lines respectively. We draw the yellow moving direction line and obtain the 2D moving direction vector $a_0^m$ in the same way utilizing $a_0^{M'}$.

## 3.3 TRAINING STRATEGY

Since using only visual prompts as input may lead the model to interpret them as object patterns rather than meaningful signals, we also incorporate language prompt inputs derived from the extracted values of visual prompts. Consequently, the base model must effectively interpret inputs from both modalities. Given the robust language understanding and visual processing capabilities of Multimodal Large Language Models (MLLMs) and inspired by their applications in prior robotic manipulation tasks (Li et al., 2024; 2023; Liu et al., 2024b; Kim et al., 2024), we have selected MLLMs as the backbone of our approach. In this section, we demonstrate how we enable MLLMs to comprehend crayon visual prompts and equip them with manipulation capabilities.

As shown in Figure 2, we design fine-tuning tasks for MLLMs by creating pairs of inputs with varying levels of information and crafting loss functions to guide the policy training. By doing so, the model can effectively execute the required low-level actions, *i.e.*, where to contact and how to

interact. Thus, when faced with novel objects, as long as these model-interpretable prompts are provided, the model can predict accurate actions, enhancing its generalization ability.

### 3.3.1 Model architecture

We utilize the LLaMa-Adapter (Zhang et al., 2023) as the backbone and adopt its training strategy, as it allows for fine-tuning only the injected adapters (Hu et al., 2021) within LLaMA (Touvron et al., 2023), alongside the multi-modal projection module, while keeping the primary parameters frozen. This approach aims to preserve the robust generalization capabilities inherent in existing MLLMs, particularly in sim-to-real transfer, while enhancing the model's ability to comprehend visual prompts and perform robotic manipulation. Specifically, when presented with an RGB image containing visual prompts $I$, we employ CLIP's visual encoder (Radford et al., 2021) to extract visual features. Simultaneously, text prompts $P$ are encoded into text features using LLaMa's pretrained tokenizer (Touvron et al., 2023). The alignment of visual and text feature representations is achieved through the multi-modal projection module.

### 3.3.2 Policy learning

We formulate the problem of pose prediction as a language modeling task, directly outputting 3D directions in textual form. Thus, we generate the ground-truth text as shown in Figure 2 to guide the training. The entire language description is shown in Appendix A.1.

**Input pairs:** To enable the model to comprehend the meanings of various crayon prompts, we design multiple pairs of visual and textual input carrying different levels of information for the model to learn from, allowing it to gradually comprehend the meaning of each colored prompt. Specifically, as illustrated in (a)-(d) in Figure 2 input.

**Training objectives:** Our model's objective is to accurately predict 3D directions based on 2D directional prompts. Therefore, we introduce the following losses to guide the policy training:

*Text Supervision Loss $\mathcal{L}_T$:* This loss ensures the effective alignment of the model's visual and linguistic input, making sure it can output the correct output pattern. To decrease the difficulty of direction regression prediction, we transform it to classification prediction by discretizing the continuous numbers in the normalized 3D direction vector into 100 discrete bins [-50,50], with each bin spanning 0.02. We supervise the output prediction with the ground-truth answer using cross-entropy loss.

*Orthogonal Loss $\mathcal{L}_O$:* The orthogonality of a rotation matrix necessitates that its components maintain orthogonal relationships between each pair of directions. However, formulating direction prediction as separate language predictions for $a_0^Z$ and $a_0^Y$ does not explicitly consider their geometric relation. To address this, we introduce a loss function that extracts the components of $a_0^Z$ and $a_0^Y$ from the model's output text and constrains them to ensure their orthogonal relationship by calculating its cosine similarity.

*Projection Loss $\mathcal{L}_P$:* 3D directional predictions should align with their corresponding 2D directional prompts when projected back to the 2D plane. To establish an explicit connection between the input 2D directional prompt and the output 3D directions, we introduce a projection loss designed to guide their correlation. Specifically, in the model's output, we extract the contact point $a_0^p$ and map it to 3D space using the depth map and camera parameters. Subsequently, leveraging the predicted 3D direction in the Z-axis $a_0^Z$, we locate 3D points along the direction with the contact point as the center. The 3D point are then projected back onto the 2D plane, and connecting with $a_0^p$ to generate predicted 2D directional vectors $a_0^{z'}$. We employ a similar approach to acquire $a_0^{y'}$ and $a_0^{m'}$, which are then supervised based on the 2D directional prompts $a_0^z$, $a_0^y$, and $a_0^m$ using cosine similarity, where $\cdot$ denotes the dot product:

$$\mathcal{L}_P = (1 - \frac{\mathbf{a_0^{z'}} \cdot \mathbf{a_0^z}}{\|\mathbf{a_0^{z'}}\|\|\mathbf{a_0^z}\|}) + (1 - \frac{\mathbf{a_0^{y'}} \cdot \mathbf{a_0^y}}{\|\mathbf{a_0^{y'}}\|\|\mathbf{a_0^y}\|}) + (1 - \frac{\mathbf{a_0^{m'}} \cdot \mathbf{a_0^m}}{\|\mathbf{a_0^{m'}}\|\|\mathbf{a_0^m}\|})$$

Note that, $\mathcal{L}_T$ and $\mathcal{L}_O$ are consistent across all input pairs, but $\mathcal{L}_P$ varies based on the input prompts. For example, if the input does not include the hint for $a_0^m$, then we will exclude the supervision for

$a_0^m$ and $a_0^{m'}$ in $\mathcal{L}_P$. After completing the training process, the model, having been trained with dynamic input patterns, can understand each color prompt and predict the required actions based on the provided prompts. The aforementioned losses are trained simultaneously under the total objective function: $\mathcal{L} = \lambda_1 * \mathcal{L}_T + \lambda_2 * \mathcal{L}_O + \lambda_3 * \mathcal{L}_P$.

## 3.4 INFERENCE AND INTERACTION

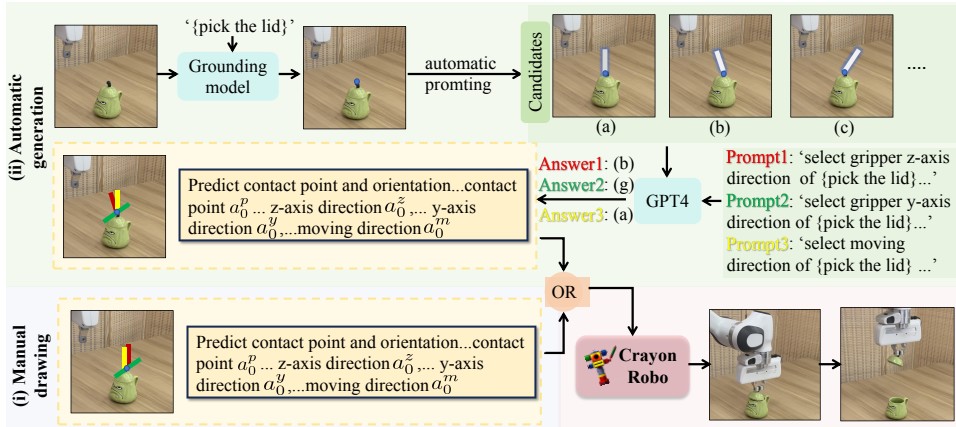

Figure 3: Illustration of model inference with input generated in different ways.

### 3.4.1 MODEL INFERENCE

During the inference stage, as shown in Figure 3(i), we allow users to draw crayon visual prompts on the image, which then serve as the visual input. The blue, red, and green prompts indicate the pose that the end effector should reach, while the yellow line represents the moving direction after contact. Meanwhile, we also provide an automated method to extract these visual prompts, as illustrated in Figure 3(ii). First, we use Grounded-DINO (Liu et al., 2023) to detect the object's bounding box and select its center, forming the blue circle. Then, we automatically generate 20 uniformly sampled 2D directional lines around the full 360-degree circle with the blue circle as the center. GPT-4 Achiam et al. (2023) is then prompted to select lines from all candidates to represent the gripper's z-axis direction, y-axis direction, and moving direction, resulting in the red, green, and yellow lines, respectively.

We observe that all long-horizon tasks can be broken down into several atomic tasks. Different atomic tasks exhibit distinct motion patterns: some require a specific moving direction $a_0^M$ to guide the subsequent movement after contact, while others depend solely on the pose for the next step. For example, *place* and *move* are simple motions that do not require additional movement after approaching the object. Once the object is placed according to the contact visual prompt, the task is considered complete. Therefore, there is no need to draw a moving prompt for these primitives.

For the *rotate* like primitive, such as rotating a button, this action typically does not involve position translation after contacting the object; instead, it rotates the last joint. Therefore, $a_0^m$ is not specified, and we can compare the prompt on the next image with the current image to determine whether the rotation should be clockwise or counterclockwise. In contrast, the some primitives require the moving direction to determine the moving action. For example, *pick* action requires moving along $a_0^M$, which is usually an upward movement. For both *pushing* and *pulling*, the robot must determine the direction in which to push or pull. Thus, for different primitive actions, the user should utilize different prompt patterns to specify the action goal, with or without moving direction prompts.

Note that while the moving direction may overlap with other directional prompts, it does not need to originate from the contact point as long as its direction is correct and clearly visible in the image. Regardless of whether it starts from the contact point, the extracted directional text prompt remains the same.

After generating visual prompt image, we automatically extract the coordinates of the contact point and the direction vectors based on the pixel RGB values, then incorporate them into the language prompt. Given the visual and language input, the model outputs the predicted action $a_0 = (a_0^p, a_0^Z, a_0^Y, a_0^M)$.

### 3.4.2 Interaction strategy

We illustrate how we enable the robot to interact with objects given the predicted action $a_0$. Specifically, $a_0^p$ is projected into 3D space $a_0^P$ utilizing depth information and camera parameters. The $a_0^Z$ and $a_0^Y$ jointly contribute to determine the rotation matrix of end-effector, facilitating the establishment of initial contact with the object. If provided with $a_0^m$, we then follow the predicted moving direction $a_0^M$ to determine the subsequent movements after contact.

For the long-horizon trajectory, we leverage a sequence of images with crayon visual prompts to serve as high-level planning, with each image representing an atomic step. By executing the atomic steps sequentially, we can complete the long-horizon tasks. The benefit of this approach revolves around breaking down the complexity of long-horizon tasks and allowing us to optimize the success rate of each atomic task to ensure the overall task's success. Meanwhile, regarding each atomic task as compositions, we can group them into arbitrary combinations, enabling the model to handle various manipulation tasks.

## 4 Experiment

### 4.1 Setup details

**Data Collection.** Following previous work (Mo et al., 2021; Li et al., 2024), we utilize SAPIEN Xiang et al. (2020) along with the PartNet-Mobility dataset to construct an interactive environment, employing the VulkanRenderer for highly efficient rasterization-based rendering. The setup uses the flying Franka Panda Gripper for both the training and testing stages. We follow the procedure in Section 3.2 to collect crayon visual prompts within the simulator. Acknowledging the inherent imprecision in real-world drawing, we introduce noise to both the visual and language prompts to bridge the realism gap during training. Details of the training and testing dataset split, as well as the input noise, can be found in Appendix A.2.

**Evaluation Metric.** We utilize the manipulation success rate to assess the effectiveness of the manipulation, calculated as the ratio of successfully manipulated samples to the total number of test samples. A successful sample is defined using a binary criterion, with success determined by thresholding the distance that the object part moves.

Table 1: Comparison of our method against baseline methods. (s) and (f) denote suction gripper and finger gripper, respectively. Bold text indicates the highest score within each end-effector type.

**Seen Categories**

| Method | C1 | C2 | C3 | C4 | C5 | C6 | C7 | C8 | C9 | C10 | C11 | C12 | C13 | C14 | C15 | C16 |
|---|---|---|---|---|---|---|---|---|---|---|---|---|---|---|---|---|
| Flowbot | 0.76 | **0.86** | 0.08 | 0.67 | 0.26 | 0.05 | 0.58 | 0.29 | 0.71 | 0.35 | 0.07 | 0.23 | 0.40 | 0.63 | 0.52 | 0.04 |
| Ours(s) | **0.89** | 0.85 | **0.67** | **0.91** | **0.87** | **0.50** | **0.92** | **0.85** | **0.87** | **0.85** | **0.54** | **0.84** | **0.85** | **0.92** | **0.85** | **0.75** |
| ManipLLM | 0.68 | 0.64 | 0.36 | 0.77 | 0.26 | 0.62 | 0.65 | 0.61 | 0.38 | 0.52 | **0.53** | 0.40 | 0.64 | 0.71 | **0.83** | 0.64 |
| Implicit3D | 0.53 | 0.58 | 0.35 | 0.55 | 0.30 | **0.66** | 0.58 | 0.51 | 0.31 | 0.41 | 0.45 | 0.34 | 0.69 | 0.54 | 0.31 | 0.43 |
| RT-Traj | 0.56 | 0.58 | 0.28 | 0.45 | 0.56 | 0.50 | 0.65 | 0.42 | 0.29 | 0.56 | 0.40 | 0.49 | 0.46 | 0.53 | 0.52 | 0.27 |
| Ours(f) | **0.71** | **0.79** | **0.58** | **0.81** | **0.79** | 0.43 | **0.81** | **0.81** | **0.75** | **0.78** | 0.50 | **0.75** | **0.76** | **0.79** | 0.81 | **0.73** |

| | **Seen Categories** | | | | | | | | | **Unseen Categories** | | | | | | |
|---|---|---|---|---|---|---|---|---|---|---|---|---|---|---|---|---|
| Method | C1 | C2 | C3 | C4 | C5 | C6 | C7 | C8 | AVG | C9 | C10 | C11 | C12 | C13 | C14 | AVG |
| Flowbot | 0.18 | 0.10 | 0.39 | 0.10 | 0.10 | 0.61 | 0.34 | 0.19 | 0.43 | 0.12 | 0.60 | 0.21 | 0.51 | 0.10 | 0.22 | 0.38 |
| Ours(s) | **0.65** | **0.90** | **0.78** | **0.79** | **0.68** | **0.91** | **0.94** | 0.46 | **0.80** | **0.47** | **0.88** | **0.83** | **0.70** | **0.76** | 0.63 | **0.79** |
| ManipLLM | 0.41 | **0.75** | 0.44 | 0.67 | 0.38 | 0.22 | 0.81 | **0.86** | 0.51 | 0.38 | 0.85 | 0.42 | 0.60 | 0.43 | **0.65** | 0.47 |
| Implicit3D | 0.27 | 0.65 | 0.20 | 0.33 | 0.45 | 0.17 | 0.80 | 0.53 | 0.55 | 0.15 | 0.41 | 0.57 | 0.39 | 0.28 | 0.52 | 0.39 |
| RT-Traj | 0.32 | 0.46 | 0.40 | 0.31 | 0.37 | 0.68 | 0.58 | 0.47 | 0.57 | 0.25 | 0.39 | 0.48 | 0.57 | 0.13 | 0.59 | 0.52 |
| Ours(f) | **0.63** | 0.67 | **0.48** | **0.78** | **0.67** | **0.74** | **0.83** | 0.44 | **0.74** | **0.46** | **0.88** | **0.82** | **0.63** | **0.61** | 0.61 | **0.72** |

## 4.2 COMPARISONS WITH BASELINES

To ensure fairness in comparison, all methods adhere to the same train and test split. The tasks involve several actions, *e.g.*, pull the drawer, open the door, lift the laptop lid, etc.

*Pure vision model* Flowbot3D (Eisner et al., 2022): It predicts motion direction on the point cloud, denoting it as 'flow'. The point with the largest flow magnitude serves as the interaction point, while the direction of the flow represents the end-effector's orientation and moving direction. It uses suction gripper as the end-effector, which is compared with ours(s) in Table. 1

*Language conditioned model* ManipLLM (Li et al., 2024): It takes the task description and object initial image, and then outputs the end-effector pose. We use its predicted contact point and rotation to contact with the object and adopt its active adaptation policy for moving.

*Vision Goal conditioned 3D model* Implicit3D (Zhong et al., 2023): It develops a manipulation policy that utilizes the transporter to detect key-points for 3D objects. By providing initial and target state point cloud of the object, keypoints are then used to determine end-effector pose for manipulation.

*Visual prompt conditioned model* RT-trajectory (Gu et al., 2023): Since the RT-Trajectory code is not publicly available at the time of this paper's submission, we replicate its method for comparisons. The replication details can be found in Appendix.A.3.

*Goal video conditioned model* AVDV (Yang et al., 2023b): The generative model is trained on our dataset, which includes tasks involving articulated objects, using the Franka arm as the end-effector. During testing, we employ Gemini-1.5 (Team et al., 2023) to evaluate the quality of the generated videos by assessing whether they successfully complete the given tasks, *e.g.,* open the door. The ratio of successfully generated tasks is 10.2%. We attribute this outcome to the complexity of tasks involving articulated objects, which typically involve movement of multiple parts. This complexity makes it challenging for the generative model to accurately capture the physical properties and dynamics of the objects. Given that the performance of the generative video-based execution policy is highly dependent on the quality of the generated videos, we believe this score reflects its overall effectiveness during execution.

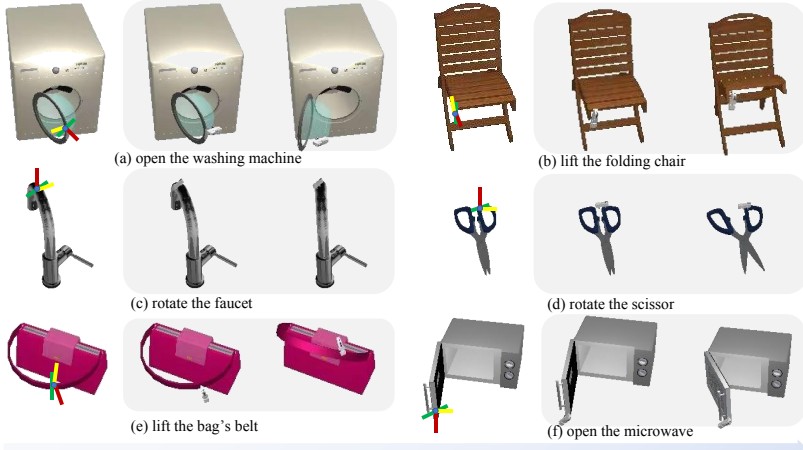

Figure 4: Visualization results in SAPIEN simulator.

In Table 1, our method demonstrates a substantial performance advantage over the baselines, showcasing its proficiency at understanding input prompts and generating precise poses. Notably, our model exhibits remarkably small performance gaps between training and testing categories, indicating its generalization by providing interpretable crayon visual prompts. To quantify the accuracy of predicted poses, we compute the cosine similarity between predicted and ground-truth directions along the z-axis and y-axis, resulting in values of 0.96 and 0.95, respectively.

Furthermore, we conduct experiments involving tasks with two atomic steps: pulling and then pushing, such as pulling a door and subsequently pushing it. Our method also results in satisfactory

performance: 0.69 and 0.68 on seen and unseen tasks. By treating atomic tasks as compositional actions, we enhance the model's generalization capabilities, enabling it to perform effectively in novel scenarios as long as the individual tasks fall within its scope of understanding.

Simulator visualizations are shown in Figure4, illustrating the visual prompt input, the robot's contact state with the object, and the final state after movement.

### 4.3 ABLATION STUDY

*Does language prompt contribute more or visual prompt?* In Table2.Ex1&Ex2, our goal is to investigate the differential effects of visual and language prompts. We train the model using images with drawn prompts as input, while the language input simply states, "Predict the contact point and directions..." without any instructional prompts. The experiment shows that relying solely on visual direction prompts leads to a noticeable decline in performance. Upon further examination, we observe that the model interprets the drawn prompts as decorative patterns on the object when there is no accompanying language description, making it difficult to understand the intended meaning of each color prompt. Additionally, we allow the model to learn from images capturing only the object, along with language containing instructional prompts. This configuration results in a small margin of difference compared to our method, which utilizes both visual and language prompts. The reduced performance can be attributed to the difficulty the model has in effectively linking the given language prompt to the relevant objects. Consequently, it becomes challenging to make accurate object-centric manipulation predictions when relying solely on language prompts. We conclude that prompts from both modalities work together to enhance the model's understanding of inputs and improve its pose prediction capabilities. This also underscores why we chose MLLMs as the backbone, due to their strong ability to process text-based input conditions effectively.

*Automatic generated prompt.* We present the results of automatically generated prompts. The results are summarized in Table 2.Ex3. Note that, since we select the center of the bounding box that is generated from GroundingDiNO as the contact point input, it may not align with the exact contact point and introduces some noise into the input positional prompt. Nonetheless, our findings demonstrate the robustness of our method in handling input inaccuracies. Even with automatically generated prompts, our approach consistently outperforms the baseline methods.

*Robustness Analysis on the Prompt* We introduce noise into the 2D directional prompt during testing while maintaining the accurate 2D contact point, in order to evaluate the model's tolerance to input disturbances. Random noise with a uniform distribution is added to the directional visual prompt, ranging from 10%, 20%, 30%, to 40% of the original directional value. The results, shown in Figure 5, indicate that with 10% and 20% noise, our method achieves performance levels comparable to those of the noise-free scenario. This demonstrates the model's ability to handle noisy inputs during testing. However, with 30% and 40% noise, performance degradation occurs as the directional values deviate significantly from their intended targets—deviations that would be unlikely to happen if drawn by a human, as they appear visibly unreasonable. Despite this, the model is still capable of successfully executing tasks, even when the directional inputs are imperfectly provided by a non-expert.

|      | VP | LP | DRW. | AUTO. | Seen | Unseen |
|------|----|----|------|-------|------|--------|
| Ex1  | ✓  | -  | ✓    | -     | 0.16 | 0.10   |
| Ex2  | -  | ✓  | ✓    | -     | 0.69 | 0.68   |
| Ex3  | ✓  | ✓  | -    | ✓     | 0.64 | 0.62   |
| Ours | ✓  | ✓  | ✓    | -     | 0.74 | 0.72   |

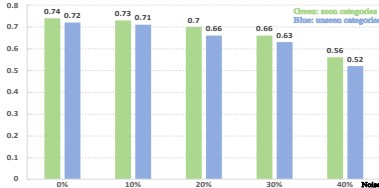

Table 2: Ablation Study.      Figure 5: Ablation on noise of input.

Ablation experiments regarding the effectiveness of each loss and effectiveness of each kind of prompt are shown in Appendix A.4.1 and Appendix A.4.2.

### 4.4 REAL-WORLD EXPERIMENT

We conduct experiments involving interaction with various real-world objects. Our setup includes a Franka Emika robotic arm equipped with a finger gripper, along with a RealSense 415 camera for

Figure 6: Illustration of executing the long-horizon tasks consisting of multiple atomic steps.

capturing RGB images and depth maps. To address the sim-to-real problem effectively, we employ two key strategies: 1) During training, we leverage the LLaMA-Adapter pretrained in the real world. We employ a fine-tuning approach that focuses solely on updating adapters to enable the model to learn new downstream manipulation tasks. This strategy allows the model to retain its robust perception abilities in the real world while acquiring novel skills. 2) When collecting data in the simulator, we employ domain randomization to enhance scenario diversity. This involves varying elements such as object part poses and camera view angles, among others, to mitigate potential sim-to-real discrepancies.

In our real-world experiment, we employ the workstation's built-in image editor to directly draw crayon visual prompts on the images, as depicted in Figure 7. We subsequently extract 2D vectors from the drawn colored lines, forming the text input. For each category, we select one or two shapes of objects and perform five trials each with a different camera view angle and object's initial pose. If the task is been completed, we consider the execution successful. As shown in Table. 3, the metric is the success rate. The results demonstrate that our proposed method can still show promising performance in real-world, even demonstrating strong generalization ability on unseen tasks. We also analyzed the failure cases: for the push button task, the excessive reactive force during button pressing prevented the robotic arm from completing the push successfully. For the slide lever task, the gripper fingers we use are too short, which sometimes prevent them from firmly grasping the lever during moving.

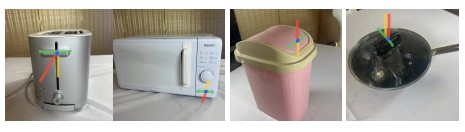

Figure 7: Real-world input demonstration.

|  | Seen Tasks |  |  | Unseen Tasks |  |
|---|---|---|---|---|---|
|  | Open the trashcan | Open micro-wave door | Push button | Lift pan lid | Slide lever |
|  | 5/5 | 4/5 | 3/5 | 5/5 | 3/5 |

Table 3: Real-world success rate.

In Figure 6, we illustrate the process of completing long-horizon tasks. More demonstrations can be found in the supplement video or at website.

## 5 CONCLUSION AND LIMITATIONS

We introduce an innovative and straightforward crayon visual prompt, which can serve as the task objective at both low-level action and high-level planning. We devise a training strategy that enables the model to comprehend the visual prompt and predict accurate contact pose along with moving direction, ensuring the reliability of task execution. Such model-interpretable prompts show promising performance on both seen and unseen objects, demonstrating its generalization ability. While our proposed approach demonstrates generalization capabilities for novel manipulation tasks, several limitations remain to be addressed. For example, the current visual prompt does not specify the dense trajectories, which requires additional caution in cases where obstacles are present along the trajectory. Therefore, future exploration could focus on using simple and straightforward visual prompts to convey more comprehensive information.

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

# A APPENDIX

## A.1 INPUT DETAILS

Due to space constraints, we outline the complete input language and ground-truth supervision for Figure 2 in this section:

Input(a): "Predict the contact point and orientation for manipulating the object. The hints in the image include the contact point with a blue dot. Specifically, the contact point is at $a_0^p$."

Input(b): "Predict the contact point and orientation for manipulating the object. The hints in the image include a blue dot for the contact point and a red line for the gripper z-axis 2D direction. Specifically, the contact point is at $a_0^p$, and the gripper z-axis 2D direction is $a_0^z$."

Input(c): "Predict the contact point and orientation for manipulating the object. The hints in the image include a blue dot for the contact point, a red line for the gripper z-axis 2D direction, and a green line for the gripper y-axis 2D direction. Specifically, the contact point is at $a_0^p$, the gripper z-axis 2D direction is $a_0^z$, and the gripper y-axis 2D direction is $a_0^y$."

Input(d): "Predict the contact point and orientation for manipulating the object. The hints in the image include a blue dot for the contact point, a red line for the gripper z-axis 2D direction, a green line for the gripper y-axis 2D direction, and a yellow line for the moving 2D direction. Specifically, the contact point is at $a_0^p$, the gripper z-axis 2D direction is $a_0^z$, the gripper y-axis 2D direction is $a_0^y$, and the gripper moving 2D direction is $a_0^m$."

Ground truth: "The contact point is at $a_0^p$, the gripper z-axis 3D direction is $a_0^{Z'}$, the gripper y-axis 3D direction is $a_0^{Y'}$, and the moving 3D direction is $a_0^{M'}$."

## A.2 DATA COLLECTION DETAILS

**Train & test split** The size of the training dataset is around 10,000. Regarding the variation between training and testing data, the specific variations can be divided into two aspects: 1) asset variation and 2) state variation.

Asset Variation: We use 20 categories from PartNet-mobility Xiang et al. (2020) for seen objects and reserve the remaining 10 categories for unseen objects to analyze whether CrayonRobo can generalize to novel categories. Specifically, we further divide the seen objects into 1,037 training shapes and 489 testing shapes, using only the training shapes to construct the training data. Thus, the shapes of the seen objects encountered during training and testing are different. For unseen categories, there are a total of 274 shapes, which are used exclusively in the testing data.

State Variation: We observe the object in the scene from an RGB-D camera with known intrinsics, mounted 4.5 to 5.5 units away from the object, facing its center. The camera is located in the upper hemisphere of the object with a random azimuth between $45°$ and $-45°$, and a random altitude between $30°$ and $60°$. We initialize the starting pose for each articulated part randomly between its rest joint state (fully closed) and any position up to half of its joint state (half-opened). In both the training and testing phases, the object is placed and captured randomly within the aforementioned scope.

**Noise on Input Prompt**: For the positional prompt, we randomly place it within a 10-pixel circle centered around the ground truth contact point. Regarding the directional prompt, we sample values with noise uniformly, allowing for a deviation of up to 20% from the original directional values.

## A.3 DETAILS OF RT-TRAJECTORY REPLICATION

Since the code for RT-Trajectory is not publicly available, we replicate its method based on the paper's description. During data collection in the simulator, we record the 3D position of the end-effector and project it onto the camera frame to create the corresponding 2D trajectory. Given that the tasks are atomic, consisting of a single step (e.g., opening a door), color grading is unnecessary. Instead, we mark the start and end positions, as well as the gripper state, by drawing blue and green circles, respectively.

We use the same backbone as in our model, the LLaMA-adapter, and fine-tune it to process both the trajectory image and the current object image. This allows the model to output the 6DoF poses required to complete the tasks. The same training and testing splits are applied, resulting in an average success rate of 0.57 on seen categories and 0.52 on unseen categories for RT-Trajectory, while our model achieves 0.74 and 0.72, respectively.

Further investigation reveals that in our replication of RT-Trajectory, while the method accurately captures the end-effector's trajectory position, the rotation estimation is not precise enough for interacting with articulated objects. Unlike tasks such as pick-and-place, where the end-effector's rotation is relatively uniform, interactions with articulated objects demand more diverse and complex rotational adjustments, making it challenging for RT-Trajectory to learn effectively. This also highlights the need to provide directional prompts for the model to interpret.

## A.4    ADDITIONAL EXPERIMENTS

Table 4: Ablation study of each component. DRA and HEU denote that the movement direction after contact follows either the drawing prompt $a_0^m$ or a rule-based selected direction, respectively.

|  | $\mathcal{L}_T$ | $\mathcal{L}_O$ | $\mathcal{L}_P$ | $a_0^p$ | $a_0^z$ | $a_0^y$ | $a_0^m$ | Motion | Seen | Unseen |
|---|---|---|---|---|---|---|---|---|---|---|
| Ex1 | ✓ | - | - | ✓ | ✓ | ✓ | ✓ | DRA. | 0.68 | 0.57 |
| Ex2 | ✓ | ✓ | - | ✓ | ✓ | ✓ | ✓ | DRA. | 0.71 | 0.70 |
| Ours | ✓ | ✓ | ✓ | ✓ | ✓ | ✓ | ✓ | DRA. | 0.74 | 0.72 |
| Ex4 | ✓ | ✓ | ✓ | ✓ | - | - | - | HEU. | 0.37 | 0.27 |
| Ex5 | ✓ | ✓ | ✓ | ✓ | ✓ | - | - | HEU. | 0.45 | 0.34 |
| Ex6 | ✓ | ✓ | ✓ | ✓ | ✓ | ✓ | - | HEU. | 0.73 | 0.70 |

### A.4.1    THE EFFECTIVENESS OF EACH LOSS.

In Ex1-Ours of Table 4, we progressively introduce each loss term during training: Ex1 involves training solely with $\mathcal{L}_T$; Ex2 combines $\mathcal{L}_T$ with $\mathcal{L}_O$; and Ours integrates $\mathcal{L}_T$ with both $\mathcal{L}_O$ and $\mathcal{L}_P$. Comparing Ex2 and Ex1, we observe that incorporating $\mathcal{L}_O$ enhances accuracy by explicitly enforcing the orthogonality constraints between the z-axis and y-axis directions. Furthermore, the addition of $\mathcal{L}_P$ in Ours results in a further accuracy improvement compared to Ex2, showing its effectiveness in capturing the correlation between 2D prompts and 3D directions.

### A.4.2    DOES EACH HINT HELP?

In Ex4-Ex6, since our model is able to handle various input patterns thanks to the proposed training strategy, we progressively introduce each hint during testing. When no $a_0^m$ is provided, we use rule-based seleted trajectories to determine the movement after contact, denoted as 'HEU' in Table 4. Beginning with Ex4, where only a 2D position prompt is provided, the model achieves impressive performance with scores of 0.37/0.27. We compare our results with using AnyGrasp (Fang et al., 2023) to predict rotation given the same pixel coordinate, which results in lower scores of 0.25/0.21. This shows even without directional prompts, our model can accurately predict poses, showcasing its ability to comprehend objects and predict appropriate poses based on positional prompts. Moreover, comparing Ex5 and Ex6, we observe that by introducing more prompts during testing, the model achieves more accurate predictions.

