# OpenReview forum: "CrayonRobo: Toward Generic Robot Manipulation via Crayon Visual Prompting"
_ICLR.cc/2025/Conference — ICLR 2025 Conference Withdrawn Submission_

### Official Review · Reviewer_N1Rd · 2024-10-22

**Soundness:** 3
**Presentation:** 2
**Contribution:** 3
**Rating:** 5
**Confidence:** 3

**Summary:**

In this paper, the authors introduce CrayonRobo which leverages simple 2D crayon visual prompts to convey both low-level actions and high-level planning for robotic manipulation tasks. The key contribution is the use of human-drawn or automatically generated crayon prompts on images, which specify the robot's end-effector pose, moving direction, and contact point in SE(3) space. The authors propose a training strategy to interpret these prompts, ensuring effective task completion through atomic step predictions. Experimental results demonstrate strong generalization to both seen and unseen manipulation tasks in simulated and real-world environments​.

**Strengths:**

1. Novel Use of Crayon Visual Prompts: While the prediction of contact points and interaction patterns is a well-explored concept, the use of crayon visual prompts as an intermediate action representation is both novel and practical. This approach avoids the common pitfalls of language ambiguity and the overly specific nature of goal images or videos.

2. Solid Experimental Setup: The paper presents a robust experimental evaluation across simulated and real-world environments. It carefully compares the method with strong baselines such as Flowbot3D and RT-Trajectory. The authors include both simulation and real-world tests along with detailed metrics, demonstrating the method's applicability.

**Weaknesses:**

1. Lack of Temporal Reasoning for Long-horizon Tasks: While the method is applied to long-horizon tasks, it lacks temporal reasoning. There is no mechanism for the model to use past actions or predict future steps to ensure consistency over the entire task.

2. Action Representation Gaps: The current formulation of action representation, $a_0$, seems inadequate for certain essential atomic actions, such as those involving joint rotations after initial contact (e.g., turning a door knob, opening a bottled water, or flipping a bottle). While these types of actions are briefly mentioned (lines 313-320), the explanation lacks depth, and it's unclear how the method handles complex manipulations that require both translation and rotation during the same task phase.

3. Writing Issues and Typos: The paper contains several minor writing issues and typos. These include the misuse of \citet vs. \citep in related works (e.g. lines 125 and 143), as well as in the method section (e.g. line 304). Additionally, typos such as "the some primitives" in line 316 need correction.

**Questions:**

1. Data Collection Efficiency: The paper describes data collection as occurring only when successful manipulations happen in random interactions. How efficient is this data collection process? Does the method require a large number of interactions before collecting enough successful examples to train the model effectively?

2. LLM/VLM Sensitivity: The system involves multiple components (e.g., GPT-4, Grounded-DINO, CLIP, and LLaMA). How robust is the overall system if one of these components fails? For instance, if Grounded-DINO selects a poor initial contact point, can the system recover through CrayonRobo’s interpretation? Are there any selection or ranking mechanisms in place to ensure robustness during evaluation?

3. Soft Orthogonality Constraint: The paper enforces the orthogonality constraint via the soft loss $L_O$. Why did the authors choose this approach instead of using differentiable hard constraints, such as Gram-Schmidt orthonormalization given $a^Z_0$ and $a^Y_0$?

4. Ambiguity in Movement Directions: Is the task prompt (e.g., “pick up the lid”) also an input to CrayonRobo? If not, how does the system handle ambiguous actions when $a_0^m$ is not provided? For example, after the robot makes contact at $a_0^P$ with the pose specified by $a^Z_0$ and $a^Y_0$, the correct direction to move (e.g., left, right, inward, or outward) is often not clear. How does the system resolve these ambiguities?

---

### Official Review · Reviewer_TM9g · 2024-10-30

**Soundness:** 3
**Presentation:** 3
**Contribution:** 3
**Rating:** 5
**Confidence:** 4

**Summary:**

The paper presents a novel approach to robotic manipulation by drawing "visual prompts" to convey task goals. Unlike natural language, which can be ambiguous, and images/videos, which can be overly specific, these simple 2D drawings explicitly indicate both low-level actions and high-level plans for robotic tasks.

**Strengths:**

- The use of crayon visual prompts offers a novel way to convey task goals, addressing issues of ambiguity and over-specification in existing methods.
- The training strategy that incorporates human-drawn prompts helps the model generalize better to unseen tasks.
- The method is evaluated in both simulation and real-world settings, providing a comprehensive assessment of its performance.
- Overall, the paper is well-written and motivated.

**Weaknesses:**

- Current approach has very limited trajectory specification, which does not allow dense predictions for precise tasks.
- The proposed training strategy might not adequately cover the diversity of real-world scenarios.
- Depending on the task's complexity, users may experience cognitive load when creating the visual prompts, which could hinder the overall usability of the approach.

**Questions:**

- As tasks grow in complexity, such as dextrous manipulation or robustness in the presence of obstacles, how does this approach scale?
- How does your approach address potential ambiguities in the visual prompts, especially in more complex tasks where multiple interpretations may exist?
- How well would this method work for tasks that require high precision such as insertion or high reactivity such as navigating in the presence of several dynamic obstacles?
- Minor edits: L143 - as as

---

### Official Review · Reviewer_CFLK · 2024-11-03

**Soundness:** 2
**Presentation:** 2
**Contribution:** 2
**Rating:** 5
**Confidence:** 4

**Summary:**

This paper proposes employing crayon visual prompts explicitly convey the task objective.

This paper trains a model that comprehends the 2D prompt and predicts contact pose along with moving direction in SE(3) space.

Experiments at scale preliminarily demonstrate its promising performance and generalization ability.

**Strengths:**

The writing and organization of this paper are good.

Extensive ablation studies have proven the effectiveness of each design.

Compared to the baselines, this paper has achieved better experimental results.

**Weaknesses:**

Why use MLLMs as the model architecture for the policy? On one hand, it could introduce significant inference latency. On the other hand, it's difficult to directly fine-tune them end-to-end. Therefore, I am wondering how the performance would be if a regular lightweight model is used for initialization? For example, using BeiT, or even a pure vision model (no LLMs, without language, end-to-end training, single task)?
In my view, current language models do not seem to be very good at making visually fine-grained prediction tasks.


Although it provides more manipulation signals compared to language, is it realistic to manually draw crayon visual prompts on images in practical applications? Can crayon visual prompts be derived through learning-based models?

Therefore, I am somewhat concerned that this method may not truly enhance the generalization of robotic manipulation, as Crayon Visual Prompting seems difficult to be generated robustly and automatically.

**Questions:**

Please see the weaknesses section.

---

> ### Author Response · Authors · 2024-11-14
>
> Thank you once again for your insightful suggestions. Compared to the previous submission, we have made significant progress based on your valuable feedback. In this version, the model no longer relies solely on manually drawn visual prompts; it can now also work with automatically generated prompts. We will continue to make further improvements in the next submission. Thank you.

---

### Official Review · Reviewer_7eZQ · 2024-11-04

**Soundness:** 3
**Presentation:** 3
**Contribution:** 3
**Rating:** 6
**Confidence:** 3

**Summary:**

This paper introduces CrayonRobo, a novel approach for robotic manipulation that uses crayon visual prompts to indicate low-level actions and high-level planning. They propose that existing methods using language, goal images, or videos for task specification have limitations, such as ambiguity in language or redundancy in images. CrayonRobo addresses these limitations by employing simple 2D colored prompts drawn on RGB images. These prompts represent the desired end-effector pose and moving direction after contact. The model is trained to understand each color prompt and predict the corresponding SE(3) contact pose and 3D movement direction, facilitating atomic task completion. For long-horizon tasks, the model utilizes a sequence of images with crayon prompts, representing a series of atomic actions. The paper highlights the effectiveness of CrayonRobo through experiments in both simulated and real-world environments, demonstrating its ability to accurately interpret visual prompts and achieve a high manipulation success rate. The results also show that CrayonRobo exhibits strong generalization ability, performing well on unseen tasks.

**Strengths:**

- Intuitive and User-Friendly Interface: The use of crayon visual prompts provides a simple and intuitive way for users to specify manipulation tasks for robots. This approach is more user-friendly compared to complex language instructions or detailed pre-defined goal images, potentially making it accessible to a wider range of users, even those without expertise.

- Effective Communication of Low-Level Actions and High-Level Planning: The crayon prompts effectively indicate both low-level action details, such as the desired end-effector pose, and high-level task planning by using a sequence of images with prompts for each atomic step. This clear communication of task objectives enhances the robot's ability to understand and execute complex manipulation tasks.

- Generalization Ability: The paper demonstrates that CrayonRobo shows strong generalization ability, performing well on unseen tasks and object categories. This suggests that the model can learn to interpret the visual prompts in a way that generalizes beyond the specific examples it was trained on.

**Weaknesses:**

Limited Trajectory Details: The current visual prompt focuses on specifying the end-effector pose and moving direction but lacks details on the complete trajectory. This could lead to challenges in environments with obstacles, as the robot may not be able to plan a collision-free path based on the provided prompts.

- Potential Scalability Limitations: While they demonstrate the effectiveness of CrayonRobo on a set of manipulation tasks, it is unclear how well the approach would scale to more complex tasks involving a larger number of objects or more intricate interactions.

- Dependence on Depth: The method relies on *accurate* depth to map the 2D contact point to the 3D space. This could limit the applicability of CrayonRobo in scenarios where acquiring reliable depth is challenging, such as in environments with reflective surfaces or poor lighting conditions.

**Questions:**

- Would you please analyze more on potential methods to incorporate detailed trajectory into the existing crayon prompt framework? For instance, would a hierarchical system where initial prompts define broad strokes of movement, followed by refined prompts for obstacle avoidance, be feasible? How might this impact the model's training and inference complexity?

- While it mentions the potential for composing atomic actions for longer tasks, the lack of experiments on complex multi-step manipulations raises concerns about scalability. How would CrayonRobo handle tasks requiring sequential tool use or fine-grained manipulation of multiple objects? Are there plans to conduct experiments on such complex scenarios to better assess the method's scalability and potential bottlenecks?

- Would you please discuss strategies for handling real-world scenarios where depth might be unreliable or unavailable? Would incorporating alternative methods like monocular depth estimation or tactile sensing be considered to enhance the robustness of contact point prediction in such cases?

---

### Official Review · Reviewer_aM8P · 2024-11-07

**Soundness:** 3
**Presentation:** 2
**Contribution:** 3
**Rating:** 5
**Confidence:** 4

**Summary:**

This paper presents an interesting approach for using VLMs on robot manipulation problems by drawing crayon-like visual prompts on RGB images. These prompts convey the axes, contact point and moving direction of the robot gripper, helping the model predict gripper poses and movements in 3D space (SE(3)). The VLM is fine-tuned to translate the colored prompts and 2D directions in text into 3D on a synthetic dataset, and show transfer to unseen object categories and real-world scenarios.

**Strengths:**

- I really like the problem formulation, decomposing a complex task sequence into atomic pick and place primitives based on contacts. This is similar in spirit to M2T2 [1], but instead of training a network from scratch, it utilizes prior knowledge from VLM pertaining.
- The evaluation protocol emphasizes on generalization to novel categories and sim-to-real transfer, which are important tests for the method's real-world applicability.

[1] Yuan, Wentao, et al. "M2t2: Multi-task masked transformer for object-centric pick and place." arXiv preprint arXiv:2311.00926 (2023).

**Weaknesses:**

- It is a little underwhelming to see that the method only translates 2D vectors in text into 3D, instead of really interpreting the visual prompts. This can be seen from the significant drop in performance in Table 2 if language (LP) is removed. Plus, the method does not completely get rid of depth maps since the contact point is still predicted in 2D. This is a weird design choice that makes me think the method does not work with 3D point prediction. It would make more impact if the VLM can auto-complete the crayon drawings, e.g. predict y-axis and moving direction based on contact point, z-axis and task prompt.
- The method relies on a lot of user inputs. The user needs to draw 1 point + 3 directions for each step. Since the model is trained with incomplete prompts (e.g. only contact point), why is this not tested in evaluation?
- It is hard to derive statistical significance from only 5 trials for the real-world evaluation, especially considering the noise from low-level systems.

**Questions:**

- How are gripper close/open determined? This could easily be incorporated in the output of VLM to make the pipeline more automated.
- Where are the numbers for long-horizon tasks like wipe the table?
- Fig. 3 should show (g) instead of (c) as candidates. Otherwise it's confusing.
- Why do actions like placing do not have a moving direction? Something that could be considered is a "pre-move" direction, since placing is often more stable with a 2-step approach -> place sequence.
- If the movement direction is perpendicular to the image plane (i.e. pointing into / away from camera center), how could it be drawn?

---

### Note · Authors · 2024-11-14

I have read and agree with the venue's withdrawal policy on behalf of myself and my co-authors.